# Platelets and Thrombotic Antiphospholipid Syndrome

**DOI:** 10.3390/jcm13030741

**Published:** 2024-01-27

**Authors:** Ibrahim Tohidi-Esfahani, Prabal Mittal, David Isenberg, Hannah Cohen, Maria Efthymiou

**Affiliations:** 1Haematology Department, Concord Repatriation General Hospital, Sydney, NSW 2139, Australia; 2Sydney Medical School, Faculty of Medicine and Health, University of Sydney, Sydney, NSW 2050, Australia; 3Department of Haematology, University College London Hospitals NHS Foundation Trust, London NW1 2BU, UK; 4Haemostasis Research Unit, Department of Haematology, University College London, London WC1E 6DD, UK; m.efthymiou@ucl.ac.uk; 5Centre for Rheumatology, Division of Medicine, University College London, London WC1E 6JF, UK

**Keywords:** antiphospholipid syndrome, antiphospholipid antibodies, thrombosis, platelets, antiplatelets

## Abstract

Antiphospholipid antibody syndrome (APS) is an autoimmune disorder characterised by thrombosis and the presence of antiphospholipid antibodies (aPL): lupus anticoagulant and/or IgG/IgM anti-β2-glycoprotein I and anticardiolipin antibodies. APS carries significant morbidity for a relatively young patient population from recurrent thrombosis in any vascular bed (arterial, venous, or microvascular), often despite current standard of care, which is anticoagulation with vitamin K antagonists (VKA). Platelets have established roles in thrombosis at any site, and platelet hyperreactivity is clearly demonstrated in the pathophysiology of APS. Together with excess thrombin generation, platelet activation and aggregation are the common end result of all the pathophysiological pathways leading to thrombosis in APS. However, antiplatelet therapies play little role in APS, reserved as a possible option of low dose aspirin in addition to VKA in arterial or refractory thrombosis. This review outlines the current evidence and mechanisms for excessive platelet activation in APS, how it plays a central role in APS-related thrombosis, what evidence for antiplatelets is available in clinical outcomes studies, and potential future avenues to define how to target platelet hyperreactivity better with minimal impact on haemostasis.

## 1. Introduction

Thrombotic antiphospholipid syndrome (APS) is a complex immune-mediated thrombotic disorder, classified by laboratory features: persistent antiphospholipid antibodies (aPL), namely lupus anticoagulant (LA) and/or IgG/IgM anti-β2-glycoprotein I (aβ2GPI) and IgG/IgM anticardiolipin antibodies (aCL), together with thrombosis [1], with thrombocytopenia recently included again as a clinical criterion [2]. APS confers significant morbidity on a relatively young population, with a mean age of 45 years at diagnosis [3]. It has an incidence of 2 to 5 per 100,000 persons/year and prevalence of approximately 50 per 100,000 [4]. In patients under 50 years of age, persistent aPL are present in 17% of patients with ischaemic stroke and approximately 10% of those with unprovoked venous thromboembolism (VTE) [5,6]. A small proportion of APS patients can develop catastrophic thrombotic crises termed as ‘catastrophic APS’ (CAPS), which carries a mortality rate of 30% despite current standard of care, triple therapy with anticoagulation, corticosteroids and plasma exchange/intravenous immunoglobulin [7]. The current standard of care for APS is primarily anticoagulation with vitamin K antagonists (VKA), with consideration for addition of an antiplatelet agent for arterial thrombosis [8,9]. High rates of recurrent thrombosis remain despite optimal anticoagulation, with one meta-analysis finding 5.4% recurrence within two years after venous thrombosis, and concerningly, 22% following arterial thrombosis [10]. Improving these poor outcomes in APS patients represents a significant area of need and including appropriate antiplatelet therapy may address this, as suggested by a recent meta-analysis of arterial thrombosis in APS [11].

Many pathophysiological processes have been shown to act in concert to cause APS-related thrombosis, including activation and recruitment of multiple cell types—endothelial cells, platelets, leukocytes—as well as the coagulation and complement systems (Figure 1). This complex interplay manifests in heterogeneous thrombosis, which can occur in any vascular bed; venous, arterial, or microvascular. While the concept of targeting the many pathophysiological processes is overwhelming and would potentially have substantial adverse effects, clinical focus has mainly been on targeting coagulation factors. However, platelets have been shown to have a significant role in thromboses within any vascular bed, raising the consideration that platelets could have a central role in APS [12,13]. Thromboses within arteries are known to be mainly driven by platelets, typically initiated by adhesion to pathologically exposed subendothelial collagen, followed by platelet activation and aggregation [14]. Venous thrombosis has traditionally been considered to be fibrin and red cell rich, with intact endothelium and a lesser role for platelets. However, more recently, in vitro models have demonstrated venous thrombus extension driven by a procoagulant subpopulation of platelets in a glycoprotein (GP)VI-dependent manner [15]. This procoagulant platelet subpopulation forms after strong stimulation, most potently with thrombin and collagen, which causes sustained raised cytosolic calcium, leading to translocation of phosphatidylserine (PS) from the inner to the outer platelet membrane, providing a negatively charged surface for coagulation factor complexes to form and generate a burst of thrombin to drive fibrin formation [16]. This platelet subpopulation is more implicated in thrombosis than haemostasis, making it an attractive target for novel therapeutics. Microvascular thrombosis is characterised by fibrin and platelet-rich thrombi, with a critical role for the procoagulant properties of platelets, and their interaction with neutrophils, now apparent [17]. 

This review explores the increased platelet activation observed in APS and the mechanisms driving it. It outlines how platelets interact with the various cells and complement factors involved in APS pathophysiology, and also assesses a possibly greater role of antiplatelets in APS management.

## 2. Platelet Activation in Antiphospholipid Syndrome

### 2.1. Platelet Activation

Platelets can be activated through a number of signalling pathways. Signalling can be initiated by either fixed agonists, such as exposed subendothelial collagen, or by soluble agonists such as thrombin. These pathways have the purpose of increasing cytosolic calcium concentration, which is a prerequisite for achieving an activated state. Platelet receptors typically involved include G-protein coupled receptors (GPCR): protease-activated receptors (PAR) 1 and 4 (thrombin receptors), adenosine diphosphate (ADP) receptors P2Y_1_ and P2Y_12_, the thromboxane receptor and the adrenoreceptor; immunoreceptor tyrosine-based activation motif (ITAM) linked receptors (ILR): GPVI and FcɣRIIa, as well as GPIbα (major ligand von Willebrand Factor-vWF). The majority of platelet signalling pathways result in phospholipase C (PLC: β and ɣ) activation, a key step for liberating calcium from organelle stores such as the endoplasmic reticulum [18]. This increased calcium drives granule release, thromboxane (TXA) production, and together with the PLC-generated diacylglycerol (DAG), activates calcium and DAG-regulated guanine nucleotide exchange factor I (CalDAG-GEFI) [19], which results in integrin αIIbβ3 changing conformation to one that can bind fibrinogen for platelet aggregation [19,20]. If this increase in calcium is sustained, typically through extracellular calcium entry, a subpopulation of platelets will become procoagulant [16].

### 2.2. The Relationship between Antiphospholipid Antibodies and Platelet Activation

There appears to be an interdependent relationship between aPL and activated platelets. As described, activated procoagulant platelets expose anionic PS on their surface. β2GPI binds to PS [21], including on activated platelet membranes [22], altering its conformational structure to allow for exposure of the antigenic epitope on domain I of β2GPI, resulting in β2GPI-specific antibody production [23,24,25]. Persistent exposure to anionic surfaces allows for sustained antigenicity and ongoing autoantibody production [26]. Thus, the presence of persistent or recurrent platelet activation may play a significant role in APS pathogenesis. Furthermore, platelets release platelet factor 4 (PF4) from their alpha granules upon activation, tetramers of which bind to β2GPI, creating β2GPI dimers [27]. Both purified β2GPI dimers and dimers within the PF4-β2GPI complex cause platelet activation, with PF4-complexed β2GPI having greater antigenicity/greater aβ2GPI binding than the non-complexed form [27,28]. Together, these studies show that platelet activation during triggering events such as surgery and infections can be a driver in APS pathogenesis and the second hit mechanism required for an APS-related thrombotic event.

Persistent low-level activation of platelets is seen in APS patients. When measuring thromboxane B2, a metabolite of TXA and a marker of platelet activation through the arachidonic acid (AA) pathway, in blood and urine, patients with APS have higher levels than controls [29,30]. In one study, there was a dose dependent increase in thromboxane B2 with increasing titres of aβ2GPI [29]. This increased platelet activation at steady state, i.e., separate from thrombosis or acute phase events, is supported by studies of other activation markers. APS patients have higher levels of circulating platelet-derived microparticles, which have procoagulant properties [31,32]. Plasma markers of platelet activation, such as soluble P-selectin and soluble CD40 ligand, and platelet activation markers CD63 expression (dense granule release) and PAC-1 binding (activated integrin α_IIb_β_3_) have also been shown to be increased compared to controls [33,34]. Linking this activation to an aPL effect, aCL complexed to β2GPI obtained from APS patients was shown to induce increased thromboxane B2 secretion from healthy donor platelets [30]. Furthermore, heat-inactivated (which removes complement and coagulation proteins) APS patient serum was seen to increase platelet-derived prothrombinase activity, implicating aPL-induced procoagulant platelet formation [35]. This was supported by another study that demonstrated greater PS exposure and prothrombinase activity following incubation with aPL, an effect that reversed the anticoagulant effect of aPL in a thrombin generation system, thus indicating that platelets are required for the procoagulant effect of aPL [36]. Finally, platelets incubated with aPL exhibited substantially more coverage and thrombi formation on collagen and subendothelium in ex vivo models than controls [37,38], giving this increased activation functional relevance.

### 2.3. Mechanisms of aPL-Induced Platelet Activation

Many studies have been published evaluating the mechanism of platelet activation in APS. While there are indirect platelet activation pathways involved through the effect of aPL on other cells complement, and coagulation factors, this section will focus on the direct aPL-dependent platelet activation (Figure 2).

One of the main platelet receptor targets of aPL is apolipoprotein E receptor 2 (ApoER2). In a murine model of thrombosis, aβ2GPI and β2GPI dimers exacerbated thrombosis, an effect lost in ApoER2 knockout mice [39]. Dimerisation of β2GPI by aPL substantially increases the phospholipid binding of β2GPI and the lupus anticoagulant activity of aPL [40,41], with β2GPI dimers mimicking the in vitro activity of aβ2GPI-β2GPI complexes [42]. Β2GPI dimers increase the adhesion of platelets to collagen and fibronectin under venous and arterial shear stress, an effect shown to be ApoER2-dependent, with binding of β2GPI dimers to ApoER2 confirmed with co-immunoprecipitation [28,43,44]. Binding of aPL/β2GPI dimers to ApoER2 has also been shown to increase platelet aggregation, TXA production, granule release, and integrin α_IIb_β_3_ activation [43,45]. The effect of the ApoER2 pathway depends on concurrent major platelet glycoprotein, GPIbα, signalling. β2GPI dimers also bind GPIbα, GPIbα and ApoER2 are complexed on the platelet surface, and inhibition of GPIbα through a variety of means abrogates the effect of aβ2GPI/β2GPI dimers on platelet activation and adhesion [43,44,45]. Both pathways appear to be required to drive platelet activation through p38 MAPK phosphorylation, as blockade of either ApoER2 or GPIbα completely reverses the aβ2GPI-induced p38 MAPK phosphorylation [45]. Indeed, p38 MAPK phosphorylation is an important part of aPL-induced platelet activation seen in other studies, increasing the potency of very low levels of platelet agonists [27,46]. GPIbα-mediated platelet activation by aPL has also been shown to involve enhanced phosphatidylinositol 3-kinase (PI3K) signalling, which is seen in aPL-induced platelet activation through toll-like receptor 2 (TLR2) as well [47,48]. 

The GPIbα pathway of platelet activation is also important with regards to the altered secretion and function of vWF seen in patients with APS. vWF is critical to the adhesion of platelets to a site of injury, binding to exposed subendothelium and capturing platelets through the binding of its A1 domain to platelet GPIbα [49]. This initial capture triggers GPIbα signalling, which results in increased cytosolic calcium and granule release [50], as well as allowing for binding of collagen to integrin α_2_β_1_ and GPVI for further platelet activation and recruitment [49]. vWF is regulated by ADAMTS13, which cleaves ultra-large multimers of vWF to reduce the degree of platelet binding. When ADAMTS13 activity is impaired, excessive platelet activation and consumption occurs, the severe form of which is the life-threatening disorder thrombotic thrombocytopenic purpura [49]. Immunoglobulin G from patients with APS has been shown to stimulate vWF release from endothelial cells ex vivo [51,52], and excess vWF levels were confirmed in vivo in APS patients [53]. There is also evidence that aPL increase the platelet-binding activity of vWF. A subset of aβ2GPI isolated from APS patients was also found to inhibit the activity of ADAMTS13 [52]. Furthermore, β2GPI can inhibit binding of platelets to the A1 domain of vWF and the presence of aβ2GPI was shown to neutralise this [54]. Primary (not associated with other autoimmune disorders) APS patients have higher levels of tyrosine nitrated β2GPI, which cannot inhibit vWF-platelet binding compared to the non-nitrated form [55]. Together, these studies demonstrate increased vWF quantity and function in APS, providing another mechanism for excess platelet activation in APS.

Multiple platelet activation pathways, through GPCR and ILR, appear upregulated in APS. The ADP receptor P2Y_12_ is overexpressed in APS patients, with concurrent reduction in downstream inhibitory molecules, cyclic AMP, and GMP [56]. The same study showed hyperreactivity of APS-patient platelets to ADP, as well as healthy donor platelets incubated with IgG from APS patients [56]. Stimulation of other GPCR pathways is also potentiated in the presence of aPL, including PAR1 and 4 and the adrenoreceptor, as well as the ILR pathways stimulated by collagen [33,57,58,59,60]. One mechanism for the hyperreactivity to agonist stimulation is the aβ2GPI-induced upregulation of the mammalian target of rapamycin (mTOR) cluster 2 (mTORC2)-AKT pathway [13]. Tang and colleagues performed RNA sequencing of APS patient-platelets and found 4399 genes differentially expressed compared to healthy controls. Gene-set expression analysis found upregulated platelet activation pathways, as well as the mTOR pathway, with the authors reporting the hyperreactivity to ADP, thrombin, and collagen in the presence of aβ2GPI reversed with disruption of the mTORC2-AKT axis. This assessment was made by using an AKT inhibitor (MK2206) and with platelet-specific *Sin1* knockout mouse models. SIN1 is a subunit of mTORC2, essential for its function. The upregulation of the mTORC2-AKT pathway caused by aβ2GPI was not inhibited by FcɣRIIa-blockade. Importantly, the study further demonstrated that SIN1 deficiency in platelets reversed the aβ2GPI-induced increased arterial, venous, and microvascular thrombosis, without prolonging tail bleeding times. This highlights the importance of platelets in all forms of thrombotic APS.

There also appears to be a role for FcɣRIIa, the ILR for the Fc portion of IgG, in excessive platelet activation seen in APS. Arvieux and colleagues [60] demonstrated that aβ2GPI potentiated subthreshold concentrations of platelet agonists ADP and adrenaline in aggregating platelets, an effect inhibited by adding F(ab’)2 fragments of the antibody or IV.3, which is an FcɣRIIa antagonist. Fab fragments alone did not have any effect [60]. This suggested that aPL bind to platelets by the Fab portion and activate platelets via the Fc portion, triggering the FcɣRIIa pathway. Other studies have supported this, with one showing β2GPI immune complexes induced thrombosis and thrombocytopenia in a transgenic mouse model with humanised FcɣRIIa [61], as well as FcɣRIIa-dependence of platelet hyperreactivity demonstrated ex vivo using both aβ2GPI and anti-prothrombin antibodies (a non-criteria aPL) [62,63]. Downstream inhibition of mTOR or knockout of CalDAG-GEFI reversed the effect of aPL [61,62], highlighting their importance in this pathway. The involvement of FcɣRIIa in enhanced platelet activation, however, may be related to the specific aPL used in the experimental model, given other studies with synthesised monoclonal antibodies have demonstrated Fab fragments to be sufficient [64], or have demonstrated no effect of IV.3 [28].

Tissue factor (TF) is an initiator of the coagulation cascade, forming thrombin and fibrin. While there is some controversy over the source and measurement of TF expression in and on platelets, particularly with potential monocyte contamination, appropriately conducted studies indicate a potential role of platelet-expressed TF in APS. Platelets from APS patients (with <1 leukocyte/10^7^ platelets), and healthy platelets incubated with aPL, have been shown to have increased TF expression [65]. aPL appear to cause IRAK phosphorylation and NF-κB activation, resulting in the increased TF expression [65]. This activity is reversed upon inhibition of platelet heparanase activity, an enzyme causing degradation of heparan sulphates [66]. Platelet levels and activity of heparanase increase in stress situations such as sepsis [67], providing another potential mechanism for the “second hit” that is often seen in APS-related thrombosis.

This work collectively provides ample evidence that, regardless of which receptor or signalling pathway is triggered, the presence of aPL leads to platelet hyperreactivity.

### 2.4. Thrombocytopenia in APS

Thrombocytopenia is commonly seen in APS, typically in 20–30% of patients [68,69,70,71]. It has been associated as an independent risk factor for recurrent thrombosis in numerous studies [68,72,73,74,75]. One study reported that development of thrombocytopenia occurred in all patients that had catastrophic APS (CAPS) episodes [76], while a large registry of CAPS patients found that ~75% had thrombocytopenia [77]. While anti-platelet antibodies can frequently be seen in APS [78,79], the exact mechanism leading to thrombocytopenia has not been clarified. One study of over 200 patients found that platelet differential width and mean platelet volume were associated with thrombosis in APS, suggesting increased platelet activation and turnover (i.e., consumption), respectively [80]. In a murine model of thrombocytopenia induced by infusion of aβ2GPI/β2GPI complexes, thrombocytopenia was demonstrated to result from platelet activation and consumption in thrombosis [61]. These studies collectively suggest that a consumptive process from excess platelet activation plays a significant role in APS-related thrombosis, particularly in the more severe manifestations.

## 3. The Prothrombotic Interaction of Platelets with Other Cells in APS

Other cell types have been implicated in the pathogenesis of APS-related thrombosis. One of the most important is endothelial cells. Normal endothelial homeostasis maintains anticoagulant properties to prevent spontaneous thrombosis and aPL have been shown to cause endothelial activation/dysfunction, with reduced production of anticoagulant factors such as nitric oxide, increased TF expression, and release of procoagulant microparticles [81]. However, some studies in primary APS patients with no previous vascular risk factors have shown no difference in many markers of endothelial activation compared to normal controls, including soluble E-selectin, soluble VCAM-1, and endothelin-1 [82,83,84]. Thus, endothelial dysfunction may not be the initiating, or parallel, mechanism for thrombosis in APS. Evidence for this concept was provided in an important study by Proulle and colleagues, which concluded that platelets were likely the initial target of aPL, going on to drive the involvement of endothelial cells [12]. In a murine cremasteric arteriole and venule laser-injury model of thrombosis in APS, they only found aβ2GPI-β2GPI complex on the platelet surface within the thrombus, but none was found on the endothelial cell surface within the thrombus [12]. In this in vivo study, aPL were able to induce enhanced endothelial (intercellular adhesion molecule-1 expression) activation, platelet activation and fibrin formation. However, when using eptifibatide to inhibit platelet aggregation specifically via integrin α_IIb_β_3_, the enhancement of endothelial activation and fibrin formation by aPL was completely reversed [12]. This work strongly implicated platelets as the key driver of APS thrombosis in this microvasculature model, and this mechanism will likely hold in platelet-rich thrombi in large artery thrombosis, although this remains to be seen. Further support for the idea is provided by a study describing excess NLRP3 levels in platelet microparticles derived from platelets stimulated by aβ2GPI-β2GPI complexes. The microparticles were internalised by endothelial cells in vitro, leading to prothrombotic endothelial pyroptosis [85].

Platelets have several co-stimulatory pathways with leukocytes that are also evident in APS. Given that APS patients have increased circulating platelet-leukocyte and platelet-monocyte aggregates [34], this cross-talk likely plays a role in APS thrombosis. Monocytes in APS have been well characterised, with demonstration of increased monocyte TF expression driven by aPL [86,87,88]. While this aPL effect can be direct, there is likely contribution from platelets. Monocyte TF expression has been shown to be driven rapidly by platelet P-selectin, expressed upon platelet activation [89]. Monocytes from APS patients express higher levels of PAR2, the inhibition of which has been shown to reduce TF expression [90]. Platelet-dependent thrombin generation seen in patients with APS can, therefore, be a key trigger for monocyte TF expression and procoagulant activity. There is a similar interplay between platelet and neutrophils. Neutrophils are being identified as important contributors to thrombosis in APS through the release of their highly prothrombotic cell contents (primarily free DNA and histones) in neutrophil extracellular traps (NETs) following activation [91,92]. APS patients have higher levels of circulating NETs, with healthy neutrophil incubation with aPL shown to result in NET release [92,93]. These released NETs promote thrombin generation in vitro, confirming their procoagulant nature [92]. The presence of anti-NET antibodies that reduce NET degradation has also been identified in APS [94]. Activated platelets cause neutrophil activation and NET release [95], while histones from NETs can induce procoagulant platelet formation [96]. NET formation is highly dependent on this interaction, with one study demonstrating loss of NET formation after platelet depletion in a sepsis model [97]. The interaction of neutrophil P-selectin glycoprotein ligand 1 (PSGL-1) with P-selectin is important in thrombosis, with PSGL-1 inhibition demonstrating attenuation of venous and arterial thrombosis in a murine model of APS [98]. Looking at the platelet-neutrophil interaction more closely, platelets aggregated to neutrophils all express P-selectin and PS in both healthy controls and APS patients; however, only the APS patient neutrophils went to also express PS, thus significantly increasing generation of thrombin by the aggregated cells [99]. This increased procoagulant activity occurred despite the patients being on low molecular weight heparin and aspirin therapy, indicating that alternative targeting of procoagulant platelets, rather than targeting aggregating platelets only with aspirin, may have an additional benefit in APS.

## 4. Platelets and Complement in APS

The presence of increased complement activation is well recognised in APS [100,101,102,103], with a possible key role in APS thrombosis. Murine model genetic knockouts or with inhibited components of the complement system, namely C3, C5 and C6, have shown reversal of aPL-enhanced thrombosis and endothelial activation [104,105,106]. Complement activation is known to be procoagulant through interaction with the coagulation system [107]. However, anticoagulation with VKA has not completely overcome recurrent thrombosis in APS, especially APS-related arterial thrombosis [10]. This problem may be due to contribution of the mutually activating interaction of complement components and platelets. Platelets are known to drive complement activation through P-selectin, which facilitates C3b binding, with subsequent generation of the complement activation product, anaphylatoxin C3a, and the terminal complement complex, C5b-9 [108]. Complement can also activate platelets and is associated with thrombosis. Several studies have demonstrated a correlation between deposition of complement activation products on platelets, most commonly C4d, in systemic lupus erythematosus (SLE) patients with or without aPL with venous and arterial thrombosis [109,110,111,112]. Deposition of complement on platelets was enhanced by aPL [109,110] and was associated with higher levels of platelet activation (granule release measured by P-selectin) and aggregation responses [110,111]. The correlation with arterial thrombosis was particularly evident with higher C4d levels deposited on healthy donor platelets following addition of aPL-containing serum from SLE patients [109]. These studies suggest a link between complement, platelet activation, and thrombosis in APS. Potential mechanisms for this link are seen in studies demonstrating that complement activation products C3a and C5b-9 can directly activate platelets. At high concentrations, C3a can induce platelet aggregation, with lower concentrations causing sensitisation of aggregation response to platelet agonist ADP [113]. In vivo studies with C3 knockout models confirmed reduced platelet aggregation and less venous and arterial thrombosis in mice [114,115], with the platelet activation mechanism subsequently demonstrated to be due to binding of C3a to the platelet C3a receptor (C3aR) [116]. Platelets treated with purified C5b-9 have a large influx of calcium, a marked increase in coagulation factor V binding, and a 10-fold increase in platelet-mediated prothrombinase activity, generating thrombin and fibrin [117,118]. C5b-9 also results in the release of platelet microparticles with prothrombinase activity [119]. Collectively, these studies demonstrate the interplay of complement and platelet activation, providing another indication that procoagulant platelets play a significant role in APS and may be a therapeutic target.

## 5. The Role of Antiplatelets in the Management of Thrombosis in APS 

While APS is a heterogeneous disease with many contributing factors to thrombosis, making a one size fits all single modality treatment unrealistic, there is ample evidence that platelets play a central role and are an attractive target. Yet, antiplatelet agents do not have a clearly defined role in APS management. This is likely due to a paucity of randomised controlled trials (RCTs) specifically assessing the impact of the different antiplatelet agents. Current guidelines only recommend low-dose aspirin (LDA) as primary prophylaxis in high-risk aPL carriers, or in combination with VKA (international normalised ratio, INR target range 2.0–3.0) as one of the secondary prophylaxis options for arterial thrombosis, which also include VKA alone at INR target range 2.0–3.0 or 3.0–4.0 [8,9]. The recommendation for venous thrombosis is use of VKA with INR target range 2.0–3.0, with consideration of the addition of LDA in refractory cases [120]. 

### 5.1. Antiplatelets as Primary Prophylaxis for Thrombosis in APS

Given that the presence of aPL clearly creates a prothrombotic tendency, primary prophylaxis in asymptomatic carriers would be ideal, especially in high-risk patients, such as those who are triple aPL positive, or with other cardiovascular risk factors. Indeed, meta-analyses evaluating LDA as primary prophylaxis found a benefit [121,122], with an individual patient data meta-analysis of ~500 patients finding a hazard ratio of 0.43 (0.25–0.75 95% CI) with LDA compared to control [122]. Most patients had higher risk aPL profiles (persistent LA, multiple aPL, or high aPL-titre positivity), resulting in recommendations for LDA being restricted to these patients. Other antiplatelets have not been evaluated.

### 5.2. Antiplatelets as Secondary Prophylaxis for Thrombosis in APS

As mentioned in the introduction, recurrent APS-related thrombosis standard-of-care treatment remains a problem, particularly in APS patients with arterial thrombosis [10]. Two RCTs with >100 patients each explored increasing the intensity of VKA (2.0–3.0 vs. 3.0–4.0 or 4.5) to reduce recurrence and failed to show improvement, with a significant increase in bleeding complications [123,124]. Patients with initial arterial thrombosis were underrepresented in both studies, however, and most of the thromboses in the higher intensity arms occurred when INR was less than 3.0. A systematic review which included these studies identified INR > 3.0 as ideal for patients with arterial or recurrent thrombosis [125]. There is far less available data regarding the benefit of incorporating single or multiple antiplatelets in secondary prophylaxis, with studies limited to arterial thrombosis. A multicentre, international study including retrospective and prospective data on 139 APS patients with initial arterial thrombosis found antiplatelet alone to be inferior to anticoagulant alone, with 37.2% of the antiplatelet group having recurrent thrombosis after median 4.24 year follow-up [126]. Combined antiplatelet and anticoagulant treatment was far superior to either agent alone. One large RCT (APASS-WARSS) compared LDA to VKA in 1770 ischaemic stroke patients tested for aPL (LA and aCL only) at baseline, demonstrating no difference in outcomes between the two treatments in aPL-positive patients (*n* = 720, 41%) [127]. However, the INR target was only 1.4–2.8 and no repeat aPL testing was routinely performed to confirm APS diagnosis, which is seen in far fewer (17% under age 50) ischaemic stroke patients [6]. As such, it is difficult to draw conclusions from this study. A small RCT of 20 patients compared LDA to VKA (INR 2.0–3.0) plus LDA and reported a statistically significant reduction in recurrent thrombosis with combination therapy [128]. A review in 2021 focussing on arterial thrombosis in APS explored the role of antiplatelets, finding that VKA plus antiplatelet was superior to standard-intensity VKA alone (RR 0.43, 95% CI 0.22–0.85) [11]. However, this was based on a single retrospective study with 34 patients in these cohorts [129]. The review also found that dual antiplatelets (LDA plus P2Y_12_ antagonist or cilostazol) were superior to LDA alone, with RR 0.29 (0.09–0.99 95% CI) [11]. While the strength of evidence is weak, it appears that combination treatments including an antiplatelet drug may be effective in APS with arterial thrombosis. 

### 5.3. How Can Platelets Be Targeted Better to Improve Outcomes in APS?

Despite clear pathophysiological research indicating the value of targeting platelets, the most studied antiplatelet treatment, LDA, cannot prevent recurrent thrombosis. Aspirin only inhibits thromboxane-driven secondary platelet aggregation [130], and does not inhibit procoagulant platelet formation [131]. As described, the ADP-P2Y_12_ axis of platelet aggregation is upregulated in APS [56]. This platelet hyperreactivity was resistant to aspirin, but responsive to P2Y_12_ antagonist ticagrelor, potentiated further by alternative antiplatelet cilostazol, a PDE3 inhibitor. This work highlights the need to better evaluate the readily available P2Y_12_ inhibitors—clopidogrel, prasugrel, and ticagrelor—in APS. 

Promising antiplatelet treatment strategies requiring further research are listed in Table 1. One antiplatelet agent of increasing interest is dipyridamole. Predominantly used in combination with aspirin for secondary stroke prevention [132], there may be potential for including it in both arterial and venous thrombosis management in patients with APS. Dipyridamole exerts antithrombotic properties, similar to cilostazol, by increasing the availability of intracellular cyclic AMP, which inhibits protein kinase A and platelet activation/aggregation [133]. Furthermore, dipyridamole has been shown to (directly and probably indirectly through platelets) reduce aPL-induced NET formation from neutrophils and increased venous thrombosis in mice, suggesting a potential role for management of APS with venous thrombosis.

Refocussing antiplatelet treatment to target procoagulant platelets may be another avenue to explore. Loss of procoagulant platelets reduces thrombosis (in mice) and has a minor impact on haemostasis in mouse models and in Scott’s syndrome, an inherited genetic disorder with loss of the gene encoding the phospholipid scramblase TMEM16F required for platelet PS exposure [134,135]. This platelet subpopulation was demonstrated to be increased by aPL, tipping the balance towards excess thrombin generation by reversing the anticoagulant effect of aPL, while preserving the procoagulant effect of activated protein C resistance [35,36]. Given its role in thrombosis at any vascular site and its interactions with other cells and complement described above, procoagulant platelets provide a potential new antiplatelet target that would minimise additional compromise of haemostasis.

Other agents with antiplatelet properties have also been shown to have utility in APS and warrant further investigation. Hydroxychloroquine, an antimalarial with great clinical utility in SLE, has been shown to reduce aPL-enhanced thrombosis in mice [136]. Subsequent in vitro studies demonstrated hydroxychloroquine reversed the platelet hyperreactivity caused by aPL [137]. Statins also appear protective in APS and have multiple mechanisms of reducing platelet activation [138,139]. Complement activation appears to be a marker of more severe APS [140], and, given the known impact it has on platelet activation, is another potential target. With the mTOR pathway, especially mTORC2-AKT, having an important role in platelet hyperreactivity, mTOR, SIN1, and AKT inhibition represents exciting potential targets that could have minimal impact on haemostasis.

## 6. Conclusions

Recurrent thrombosis in APS patients represents an area of unmet need and is the leading cause of mortality in this patient population. Platelets play a central role in thrombosis and there is extensive research confirming that platelets are not only key players in APS-related thrombosis, but also have a role in the formation and persistence of aPL. Much further research is needed to delineate the best way to apply antiplatelet agents in APS. Several established antiplatelet agents are available for further evaluation, either in combination with aspirin or VKA. There are also exciting novel approaches targeting the platelet-neutrophil interactions and NETs, and the mTORC2-AKT axis, as well as the potential for utilising complement activation as a biomarker for patients who would benefit from complement inhibition to reduce platelet activation. Finally, as procoagulant platelets are the prothrombotic product of many of the pathophysiological processes in APS, they represent an important new specific antiplatelet target.

## Figures and Tables

**Figure 1 jcm-13-00741-f001:**
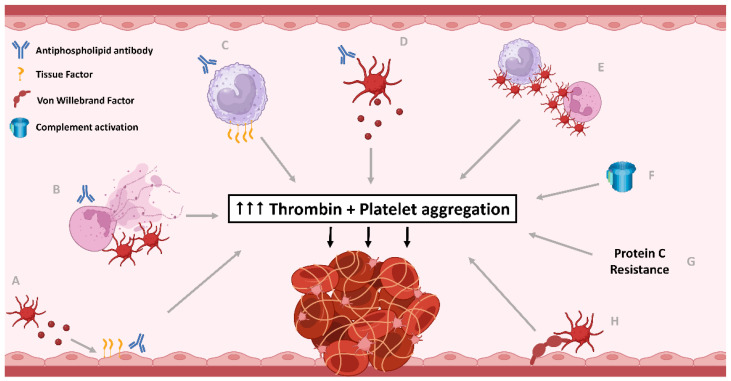
The central role of excess thrombin generation and platelet activation in pathophysiological mechanisms contributing to thrombosis in antiphospholipid syndrome (APS). Antiphospholipid antibodies (aPL) have impact on many different cells and pathways leading to thrombosis. (**A**) Endothelial cells are directly impacted by aPL causing increased tissue factor (TF) expression, as well as indirectly with aPL-related platelet microparticle-induced endothelial pyroptosis, resulting in thrombin generation. (**B**) aPL induce neutrophils to release extracellular traps (NETs), as well as reduce NETs clearance and activate platelets which also lead to NET formation, which in turn activates platelets. Neutrophils also expose phosphatidylserine (PS) to increase thrombin generation. (**C**) aPL lead to increased tissue factor expression by monocytes, causing increased thrombin generation. (**D**) aPL induce platelet activation, aggregation, and PS exposure, releasing platelet microparticles, generating bursts of thrombin for fibrin clot stabilisation. (**E**) aPL-related platelet and leukocyte activation leads to platelet-leukocyte aggregate formation, which potentiates further platelet activation and thrombin generation. (**F**) The classical and alternative pathways of complement activation are upregulated in APS, with some evidence of lectin pathway also being involved. Complement activation contributes to platelet activation and thrombin generation. (**G**) Protein C cleaves and inactivates coagulation factors V and VIII, necessary for thrombin generation. Protein C resistance is evident in APS, with anti-Protein C antibodies in APS observed in some studies. (**H**) The quantity and activity of von Willebrand Factor (vWF) is increased in APS, resulting in increased platelet adhesion, activation, and aggregation. Created with Biorender.com.

**Figure 2 jcm-13-00741-f002:**
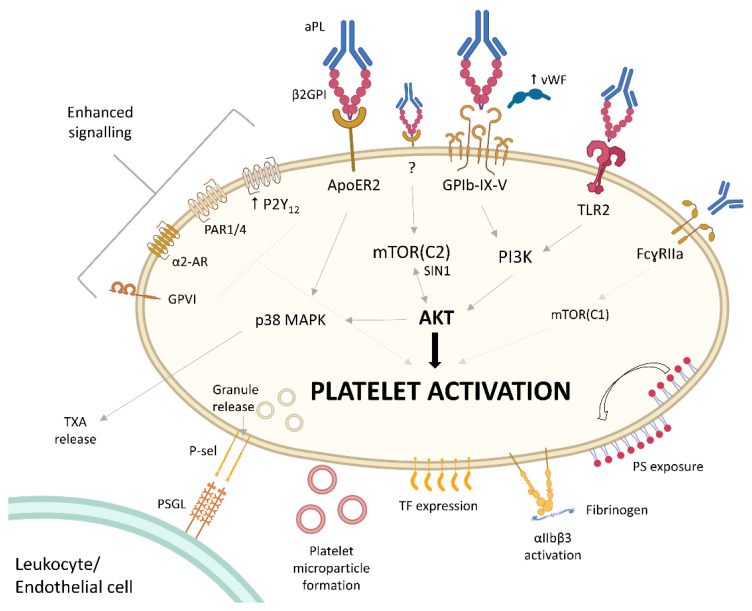
Platelets exhibit hyperreactivity through many pathways in antiphospholipid syndrome (APS). A large number of platelet receptor agonist stimulation pathways result in greater platelet activation in APS with lower concentrations of agonist. This is due to the “priming” by antiphospholipid antibodies (aPL) complexed to dimerised open-conformation β2 glycoprotein binding to Apolipoprotein E receptor 2 (ApoER2), glycoprotein (GP)Ib-IX-V receptor, and toll-like receptor 2 (TLR2). There is evidence that the Fc portion of some aPL can also prime platelets to hyperreactivity through the FcɣRIIa receptor and mTOR cluster 1 (mTORC1) signalling. The activity of AKT appears central to many of the priming pathways, most critically with the mTORC2-SIN1-AKT axis, although the upstream receptor for this pathway (shown as “?”) has not yet been identified. Hyperreactivity has been identified with ADP agonist stimulation through the P2Y_12_ receptor, which is overexpressed in APS patient platelets. Protease-Activated Receptor (PAR) 1 and 4 signalling triggered by thrombin, α2-adrenoreceptor (AR) stimulated by adrenaline, and collagen signalling (presumably through GPVI receptor and α2β1 integrin) have all been shown to be hyperreactive. The platelet activation resulting from these pathways are illustrated. TXA—thromboxane, P-sel—P-selectin, TF—tissue factor, PS—phosphatidylserine. Created with Biorender.com.

**Table 1 jcm-13-00741-t001:** Potential therapeutic strategies against platelet hyperreactivity for further research in thrombotic antiphospholipid syndrome.

Therapeutic Strategy	Available Agents	Potential Benefits/Limitations
Inhibiting P2Y_12_ receptor	Clopidogrel, prasugrel, ticagrelor, etc.	**Benefits:**
		Overexpression of P2Y_12_ in APS and associated platelet hyperreactivity Pre-clinical studies demonstrate ticagrelor can reverse platelet hyperreactivity to ADP in APS Known safety profile and clinical use together with anticoagulation
		**Limitations:**
		Increased bleeding risk, especially in combination with current standard therapy
Increasing cyclic AMP	Cilastazol, dipyridamole	**Benefits:**
		Can reverse downregulation of cyclic AMP seen in APS platelets Known safety profile
		**Limitations:**
		Increased bleeding risk, especially in combination with current standard therapy
Reducing procoagulant platelet formation	Ciclosporin, acetazolamide	**Benefits:**
		Procoagulant platelets predominantly involved in thrombosis, less impact on haemostasis Procoagulant platelets are downstream of many of the pathological processes in APS Will block the platelet-derived thrombin generation induced by aPL
		**Limitations:**
		Current agents have many off-target adverse effects that limit use
mTOR inhibition	Everolimus, sirolimus	**Benefits:**
		Preclinical studies have demonstrated reduction in platelet hyperreactivity downstream of FcɣRIIa signalling induced by aPL Will inhibit mTOR-mediated endothelial activation in APS as well No impact on haemostasis
		**Limitations:**
		Current agents have many off-target adverse effects that limit use
Inhibition of mTORC2 (SIN1)-AKT axis	?	**Benefits:**
		Mouse models have demonstrated reversal of platelet hyperreactivity and thrombosis in any vascular bed induced by aPL No prolongation of tail bleeding time in mouse model with SIN1 deficiency, so appears to be thrombosis-specific Could potentially target AKT upstream with available PI3K inhibitors
		**Limitations:**
		No specific available agents with known safety profiles in humans. AKT inhibitors and any developed agents have many off-target adverse effects that limit use
Reducing Neutrophil Extracellular Traps (NETs)	Dipyridamole, ?	**Benefits:**
		Excess NETs formation and impaired clearance in APS with prothrombotic effect Dipyridamole has a known safety profile Potential for repurposing agents that could reduce NETs, e.g., crizanlizumab (blocks P-selectin-PSGL interaction required for platelet-neutrophil interaction, used in sickle cell disease)
		**Limitations:**
		Increased bleeding risk with dipyridamole Direct NETs inhibition will likely compromise normal response to pathogens
Inhibiting excessive complement activation	Eculizumab (C5), ravalizumab (C5), sutimlimab (C1s), pegcetacoplan (C3), etc.	**Benefits:**
		Complement activation plays a key role in APS thrombosis and can induce procoagulant platelets Known safety profiles and clinical experience Would not impact haemostasis
		**Limitations:**
		Current agents have many off-target adverse effects that limit use

## Data Availability

Not applicable.

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
