# Peer review of "Platelets and Thrombotic Antiphospholipid Syndrome"

_jcm, 2024, doi:10.3390/jcm13030741_

Round 1

Reviewer 1 Report

Comments and Suggestions for Authors

The authors addressed the issue of thrombocytes in thrombotic APS.

I do not have any consideration or any other comment about a specific gap. I am thrilled by reading the manuscript. In my honest opinion the article ticks all the boxes.

Regarding methodically issues as far as a review article. I do not have any concerns about the methods used in the article.

Conclusion and discussion part is nearly answers all the questions.

References are enough and appropriate for this review.

Figures are well designed.

Author Response

Response: Thank you for your kind comments and feedback.

Reviewer 2 Report

Comments and Suggestions for Authors

This is a nice review on the current knowledge about the role of platelets in the antiphospholipid syndrome (APS). As is commonly seen with narrative reviews, there appears to be some (acceptable) bias towards the author’s opinion and preferences.

I have some minor comments:

Page 1: In 2023 new classification criteria for APS have been published. Different from the Sidney criteria cited, the new criteria include thrombocytopenia again, which underscores the relevance of platelets for APS.

Page 3, paragraph 2.2: „Persistent exposure to anionic surfaces allows for sustained antigenicity and ongoing autoantibody production.25” While this is an interesting hypothesis, the authors should take into account that stimulation of T-cells by the complex of b2GPI/PS as described in ref. 25 apparently depends on uptake of the complex by monocytes with subsequent antigen-presentation. Are there any indications that a similar process is involved with b2GPI on the platelet surface?

Page 5. Inhibition of anticoagulant function of b2GPI by anti-b2GPI is commonly discussed as a potential procoagulant effect of anti-b2GPI. It should be noted though that there is no evidence that genetic b2GPI deficiency induces thrombophilia.  

Page 6: In the paragraph on platelet tissue factor (TF) the authors should discuss the controversy regarding TF expression by platelets and the methodological difficulties to prove this.

Page 9: APASS-WARSS was not a study in APS patients, but in stroke patients. The investigators analyzed, whether aPL affected the risk for recurrent events, while in the main study (WARSS) treatment with aspirin was compared to warfarin. Accordingly, it is not surprising that most of the patients did not fulfil the APS criteria. In fact, 59% were aPL negative. Intriguingly, aPL positivity was not associated with a higher rate of vascular events during follow-up. This was independent of the treatment modality (VKA vs. aspirin).

Author Response

We would like to thank you for your most helpful and constructive comments on this manuscript. Our responses to the specific comments (italicized) are detailed below.

This is a nice review on the current knowledge about the role of platelets in the antiphospholipid syndrome (APS). As is commonly seen with narrative reviews, there appears to be some (acceptable) bias towards the author’s opinion and preferences.

I have some minor comments:

Page 1: In 2023 new classification criteria for APS have been published. Different from the Sidney criteria cited, the new criteria include thrombocytopenia again, which underscores the relevance of platelets for APS.

Response: Thank you very much for highlighting this. We have now altered the opening paragraph in the introduction to include this and have referenced the ACR/EULAR 2023 criteria.

“Thrombotic antiphospholipid syndrome (APS) is a complex immune-mediated thrombotic disorder, classified by laboratory features: persistent antiphospholipid antibodies (aPL), namely lupus anticoagulant (LA) and/or IgG/IgM anti-β2-glycoprotein I (aβ2GPI) and IgG/IgM anticardiolipin antibodies (aCL), together with thrombosis,1 with thrombocytopenia recently included again as a clinical criterion.2

Page 3, paragraph 2.2: „Persistent exposure to anionic surfaces allows for sustained antigenicity and ongoing autoantibody production.25” While this is an interesting hypothesis, the authors should take into account that stimulation of T-cells by the complex of b2GPI/PS as described in ref. 25 apparently depends on uptake of the complex by monocytes with subsequent antigen-presentation. Are there any indications that a similar process is involved with b2GPI on the platelet surface?

Response: As the reviewer has stated, Yamaguchi et al (Blood 2007) demonstrated that the T-cell stimulation response to β2GPI bound to synthetically derived phosphatidylserine (PS) liposomes in the presence of APS plasma/aPL was monocyte-dependent. The authors assessed in vivo substitutes for this synthetic PS liposome, testing oxLDL and activated platelets (as a source of PS-exposing platelets and platelet microparticles), with reproducible T-cell stimulation. Based on these findings, we stated “Thus, the presence of persistent or recurrent platelet activation may play a significant role in APS pathogenesis.”

Page 5. Inhibition of anticoagulant function of b2GPI by anti-b2GPI is commonly discussed as a potential procoagulant effect of anti-b2GPI. It should be noted though that there is no evidence that genetic b2GPI deficiency induces thrombophilia.  

Response: Thank you for the comment. We agree that inhibition of β2GPI anticoagulant function by anti-β2GPI is commonly discussed as a potential contributing factor for the procoagulant state seen in APS. However, given the complexity with seemingly contradictory evidence we feel that further details on the matter and expansion would be outside the scope of the current review that focuses on platelets.  

Page 6: In the paragraph on platelet tissue factor (TF) the authors should discuss the controversy regarding TF expression by platelets and the methodological difficulties to prove this.

Response: Thank you for the comment. We have now altered the text to address this, as below:

“Tissue factor (TF) is an initiator of the coagulation cascade, forming thrombin and fibrin. While there is some controversy over the source and measurement of TF expression in and on platelets, particularly with potential monocyte contamination, appropriately conducted studies indicate a potential role of platelet-expressed TF in APS. Platelets from APS patients (with <1 leukocyte/107 platelets), and healthy platelets incubated with aPL, have been shown to have increased TF expression.65

Page 9: APASS-WARSS was not a study in APS patients, but in stroke patients. The investigators analyzed, whether aPL affected the risk for recurrent events, while in the main study (WARSS) treatment with aspirin was compared to warfarin. Accordingly, it is not surprising that most of the patients did not fulfil the APS criteria. In fact, 59% were aPL negative. Intriguingly, aPL positivity was not associated with a higher rate of vascular events during follow-up. This was independent of the treatment modality (VKA vs. aspirin).

Response: Thank you for this comment. The text of the paragraph has been corrected, as below, to provide clarity for the reader.

“One large RCT (APASS-WARSS) compared LDA to VKA in 1770 ischaemic stroke patients tested for aPL (LA and aCL only) at baseline, demonstrating no difference in outcomes between the two treatments in aPL-positive patients (n=720, 41%).127 However, the INR target was only 1.4-2.8 and no repeat aPL testing was routinely performed to confirm APS diagnosis, which is seen in far fewer (17% under age 50) ischaemic stroke patients.6 As such, it is difficult to draw conclusions from this study.”